# The Relationship between Dietary Patterns and High Blood Glucose among Adults Based on Structural Equation Modelling

**DOI:** 10.3390/nu14194111

**Published:** 2022-10-03

**Authors:** Yuanyuan Wang, Wei Xie, Ting Tian, Jingxian Zhang, Qianrang Zhu, Da Pan, Dengfeng Xu, Yifei Lu, Guiju Sun, Yue Dai

**Affiliations:** 1Key Laboratory of Environmental Medicine and Engineering of Ministry of Education, Department of Nutrition and Food Hygiene, School of Public Health, Southeast University, Nanjing 210009, China; 2Institute of Food Safety and Assessment, Jiangsu Provincial Center for Disease Control and Prevention, Nanjing 210009, China

**Keywords:** dietary pattern, high blood glucose, structural equation modelling

## Abstract

The aim of this study was to examine the association between dietary patterns and high blood glucose in Jiangsu province of China by using structural equation modelling (SEqM). **Methods:** Participants in this cross-sectional study were recruited through the 2015 Chinese Adult Chronic Disease and Nutrition Surveillance Program in Jiangsu province using a multistage stratified cluster random sampling method. Dietary patterns were defined by exploratory factor analysis (EFA). Confirmatory factor analysis (CFA) was used to test the fitness of EFA. SEqM was used to investigate the association between dietary patterns and high blood glucose. **Results:** After exclusion, 3137 participants with complete information were analysed for this study. The prevalence of high blood glucose was 9.3% and 8.1% in males and females, respectively. Two dietary patterns: the modern dietary pattern (i.e., high in red meats and its products, vegetables, seafood, condiments, fungi and algae, main grains and poultry; low in other grains, tubers and preserves), and the fruit–milk dietary pattern (i.e., high in milk and its products, fruits, eggs, nuts and seeds and pastry snacks, but low in vegetable oils) were established. Modern dietary pattern was found to be positively associated with high blood glucose in adults in Jiangsu province (multivariate logistic regression: OR = 1.561, 95% CI: 1.025~2.379; SEqM: β = 0.127, *p* < 0.05). **Conclusion:** The modern dietary pattern—high intake of red meats—was significantly associated with high blood glucose among adults in Jiangsu province of China, while the fruit–milk dietary pattern was not significantly associated with high blood glucose.

## 1. Introduction

Diabetes mellitus (DM) is a chronic metabolic disease characterised by elevated blood glucose levels. As the disease progresses, it can further damage the heart, eyes and kidneys [1,2,3,4]. There are approximately 536.6 million people living with diabetes worldwide, and 6.7 million people die from it each year [5]. In the United States, approximately 32.2 million adults have diabetes, and 36.3 million are expected to have diabetes in 2045 [5]. In China, diabetes is considered a major health issue, with prevalence significantly increasing among adults 18 years and older, rising from 9.7% in 2012 to 11.9% in 2018 [6]. Diabetes is susceptible to unhealthy lifestyles, such as smoking, alcohol consumption and unhealthy eating habits [7,8].

Over the past few decades, studies have shown that diet acted as a major factor in the development of DM. Epidemiological studies suggested that approximately 80% of DM can be prevented through healthy dietary habits such as regular consumption of fruits and vegetables and reduced intake of saturated fat, sodium and sugar-sweetened drinks [9,10]. In a meta-analysis, vitamin D supplementation reduced the risk of type 2 diabetes (T2DM) and increased the rate of return to normal blood glucose in individuals with prodromal DM [11]. However, because of the dietary complexity of different populations and the potential of food–food or food–component interactions, it could be difficult to evaluate the effect of a single or a few foods or nutrients on DM [12,13]. One study has shown that the western dietary pattern, as determined by the Gaussian graphical models, had a positive association with the risk of T2DM in women [14]. In addition, the Mediterranean diet may reduce the risk of cardiovascular disease (CVD) in patients with DM [15].

A variety of studies on analysing dietary pattern methods have emerged in recent years [13,16]. Among them, structural equation modelling (SEqM) is an appropriate approach to statistics that merges the methods of factor analysis and path analysis to determine the direct and indirect correlations between potential and observed variables. It can take both errors and individual differences into account [17,18].

To our knowledge, there are no studies explaining the association of direct and indirect associations with high blood glucose as well as socio-demographics in Jiangsu province, China. Therefore, the aims of this study were as follows: (i) to determine the final dietary pattern by exploratory factor analysis (EFA) and confirmatory factor analysis (CFA), and (ii) to examine the effect of dietary patterns on high blood glucose among adults in Jiangsu province, China.

## 2. Participants and Methods

### 2.1. Study Population

The China Adult Chronic Disease and Nutrition Surveillance Project (2015) in Jiangsu province covered thirteen surveillance sites, including Qinhuai, Chongan, Yunlong, Wujin, Wuzhong, Zhangjiagang, Rugao, Donghai, Jinhu, Xiangshui, Hanjiang, Jingkou and Jiangyan. Recruitment of participants using multistage stratified cluster random sampling methods: ① at each surveillance site, three streets/towns were randomly selected by using a cluster sampling method proportional to the population size; ② two further villages/communities were randomly selected in each street/township using a cluster sampling method; ③ in each village/neighbourhood, one village group was selected by using a simple random sampling method (at least 60 households); ④ 45 households were randomly selected, and all residents in the households were enrolled in the survey. After screening, as shown in Figure 1, a total of 3137 participants (54.8% female, *n* = 1718) aged ≥18 years old with complete 3-day and 24 h-dietary recall data were included in this study.

### 2.2. Anthropometric Measurement

All participants were asked to wear light clothing and no shoes while taking anthropometric measurements. The TANITA HD-390 electronic weight scale (Dongwan, China) was used for weight measurement. Height measurement was performed using a TZG sit height gauge (Wuxi, China). All measurements were carried out twice to ensure the stability of the measurement results [16]. The body mass index (BMI) standard for the Chinese population was used to define the BMI classification for this study [17]. Central obesity was defined as ≥90 cm waist circumference in men and ≥85 cm in women.

### 2.3. Biochemical Indicator

Participants’ blood was collected early in the morning (8–12 h fasting). Oral confirmation of fasting was obtained from the participant prior to the blood sample collection. Then, the blood was centrifuged and fasting blood glucose (FBG) was measured using the glucose oxidase method. High blood glucose was defined as: (1) self-reported diabetes or diabetes that has been diagnosed and treated by doctors (including herbs, western medicine and insulin injections); (2) FBG ≥ 7.0 mmol/L.

### 2.4. Dietary Assessment

We used a 3-day intake of various consumed foods, including alcohol consumption, various condiments, etc., to assess the actual daily intake of individuals. The 3-day and 24-h dietary recall and food weighing method were the keys to obtaining the individual dietary intake. Professionally trained investigators recorded the foods participants had eaten in the past 24 h during the first home visit and taught them how to record their food intake. All food data were obtained through face-to-face interviews. Participants were required to record their food consumption for 3 consecutive days (including 2 weekdays and 1 weekend day). During the survey, food models and household measurement tools were used to help participants estimate their portion sizes. The collected foods were then combined into food groups according to the Chinese food composition table (2002). Using the Chinese Dietary Guidelines and combined with the dietary characteristics of the Jiangsu population, we classified the food into 22 food groups, as shown in Appendix A.

### 2.5. Dietary Pattern Analysis

For dietary pattern analysis, we mainly used factor analysis for exploration. Factor analysis includes two parts: confirmatory factor analysis and exploratory factor analysis. In this study, EFA was first used to summarise the main dietary structure of the study population. The Kaiser–Meyer–Olkin index (KMO) and Bartlett’s spherical test suggested that the data were suited for EFA. Shared factors were extracted using principal component analysis, and their number was determined based on the eigenvalues >1.3, scree plot and interpretability of derived factors. The maximum variance orthogonal rotation method was used at the same time to make each common factor more obvious and professional. The absolute value of the factor loadings >0.25 was used to determine the main food composition of each common factor. After that, we put the major foods with absolute values of factor loadings >0.25 into the validation factor analysis to determine whether the dietary pattern was in compliance.

### 2.6. Structural Equation Modelling

Structural equation modelling (SEqM), also known as covariance structural modelling, was developed by Joreskog in the 1970s [19]. SEqM consists of factor analysis and path analysis. Using this method, the acceptability of the theoretical model under specific factors can be tested. CFA was used to test the suitability of dietary patterns determined by EFA. SEqM uses a similar approach to through-path analysis to investigate the structural relationships between latent variables, and the regression path coefficients reflect the degree of correlation between latent variables. In this study, we constructed the SEqM to investigate the relationship between dietary pattern and high blood glucose. Goodness-of-fit index (GFI), adjusted comparative fit index (ACFI) ≥ 0.90, parsimonious goodness-of-fit index (PGFI), parsimonious baseline fit index (PNFI) ≥ 0.50 and root mean square error of approximation (RMSEA) ≤ 0.08 were used to confirm the degree of model fit.

### 2.7. Statistical Analysis

The mean ± standard deviation (mean ± SD) of continuous variables and frequencies of categorical variables were used to represent the distribution of general characteristics. EFA was used to identify the dietary patterns of the individuals, and the factor scores were divided into quartiles for further analysis. CFA allowed further determination of the suitability of EFA. Multivariate logistic regression analysis was used to calculate the OR and 95% CI for high blood glucose in each quartile of the factor scores. Structural equation modelling was used to investigate the correlation and degree of correlation between dietary patterns obtained from factor analysis and high blood glucose.

SAS 9.4 (Cary, NC, USA), IBM SPSS 26.0 (Armonk, NY, USA) and Origin (2021, Northampton, MA, USA) data analysis and plotting software were used for data management and statistical analysis. All statistical tests were two-sided, and differences were considered statistically significant when *p* < 0.05.

## 3. Results

### 3.1. Basic Information of Participants

Table 1 shows the basic information for the different gender groups. After exclusion, 3137 participants (54.4% male, *n* = 1708) with complete data were included in this study. The average age and energy intake of men were significantly higher than that of women (*p* < 0.001). The mean blood glucose level was higher in males than females (male = 5.5 ± 1.4 mmol/L vs. female = 5.4 ± 1.3 mmol/L, *p* < 0.05). The prevalence of high blood glucose in men was not significantly different compared to women (male = 9.3% vs. female = 8.1%, *p* > 0.05). Among participants, 45.7% (*n* = 653) males had smoking behaviour, which was significantly higher than females (1.4%) (*p* < 0.001).

### 3.2. Determination of Dietary Patterns

Figure 2 and Appendix A illustrate the dietary patterns identified by the EFA. Factor loading and interpretability were used to explain the two dietary patterns generated, namely the “modern dietary pattern” (Pattern I) and the “fruit and milk dietary pattern”. Then, we put food groupings with higher factor loadings from these two dietary patterns into the CFA model (Figure 3). Ultimately, we found that the modern dietary pattern was dominated by red meats and their products, fresh vegetables, seafood, condiments, whole cereals, main cereals, poultry, tubers and starches and their products and fungi and algae. The fruit–milk dietary pattern was dominated by nuts and seeds, fruits, eggs, milk and its products, pastry snacks and vegetable oils.

### 3.3. Analysis of the Relationship between Dietary Patterns and High Blood Glucose by Multivariate Logistic Regression

Table 2 shows the association between dietary pattern and high blood glucose in Jiangsu adults by using multivariate logistic regression modelling; the results suggest that the high intake of modern dietary patterns increased adults’ risk of high blood glucose (composed of Q1, Q3~Q4 OR = 1.566 and 1.561, 95% CI: 1.063~2.308 and 1.025~2.379, respectively, *p* < 0.05) and showed a trend toward elevation with increasing intake (*P*_trend_ < 0.05). However, the fruit–milk dietary pattern had no significant association with high blood glucose (*p* > 0.05).

### 3.4. Structural Model

Figure 4 shows the SEqM plot with standardised estimates of the relationship between dietary patterns, demographic characteristics and high blood glucose. The final SEqM model was obtained by increasing residual correlations and modification indices (as shown in Figure 4 and Table 3). The goodness-of-fit indices of the final model indicated an acceptable fit (RMSEA = 0.068, GFI = 0.913, ACFI = 0.891, PGFI = 0.727). The modern dietary pattern was positively associated with the risk of high blood glucose among adults in Jiangsu province of China. (β = 0.127, *p* < 0.001). 

## 4. Discussion

Diabetes mellitus is one of the most prominent risk factors affecting the health of the population [20]. The risk of high blood glucose among adults is on the rise in Jiangsu province, China [21]. It is suggested that we should be on alert to the further risk of high blood glucose in the region. Diet, as a controllable lifestyle, has been shown to significantly influence the development of DM [22]. In our study, two dietary patterns were identified by exploratory factor analysis and confirmatory factor analysis: modern dietary pattern and fruit–milk dietary pattern. Multivariate logistic regression and SEqM analysis revealed that the modern dietary pattern was positively associated with high blood glucose among adults in Jiangsu province, China, while the fruit–milk dietary pattern was not significantly associated with high blood glucose.

The modern dietary pattern, which was rich in red meats and its products, vegetables, seafood, condiments, fungi and algae, main grains and poultry, but was low in whole grains and tubers and preserves and was significantly associated with high blood glucose in adults. It is similar to the modern dietary pattern obtained from the China Health and Nutrition Survey (CHNS) [23,24]. The survey showed that children and adolescents aged 6–14 years and the elderly aged 60 years and older who adhere to modern dietary patterns rich in saturated fat and cholesterol are at increased risk of obesity. Obesity, in turn, is an important risk factor for diabetes [25]. Besides, we believe that the positive association between modern dietary patterns and the risk of type 2 diabetes might be partly attributed to unhealthy dietary components, such as red meats and their products and main grains. First, red meat and processed meat products, which are rich in saturated fat, have been found to be significantly and positively associated with chronic diseases such as diabetes [26,27]. In this study, our analysis classified red meat and processed meat products as a food group. Their factor loadings were first in the modern dietary pattern, indicating a high intake of red and processed meat in people who prefer the modern dietary pattern. A meta-analysis found that consuming an additional 100 g of red meat per day increased the risk of developing T2DM, while consuming 50 g and more of processed meat products per day increased the risk of T2DM by 30% [10]. Excessive intake of red meat products may lead to the overabsorption of heme iron [28]. Internal iron overload may promote insulin resistance and increase the risk of T2DM [29]. Second, excessive intake of main grains and low intake of whole grains is another major feature of this dietary pattern. In our study, main grains refer to refined rice products and wheat products. There is a general consensus that people with or at risk of type 2 diabetes should avoid carbohydrate-rich foods [30]. A systematic review and meta-analysis indicated that consuming 200–400 g of refined grains per day may increase the risk of T2DM by 6−14% [10]. In another one of our studies, we also found that excess intake of refined carbohydrates could promote elevated blood glucose [31]. Meanwhile, it is important to pay attention to the quantity of carbohydrates as well as their source and quality. Numerous studies have found that less processed whole grain foods could improve blood glucose measurements in adults with type 2 diabetes more than the same number of refined grains [32,33]. The potential reason for it is that whole grains are likely to be digested by microbiota in the colon into short-chain fatty acids, which are absorbed without altering circulating blood glucose levels [34,35]. Results from the latest prospective cohort study also showed that participants with high whole grain intakes had a 29% lower incidence of type 2 diabetes, suggesting further support for the current recommendation to increase whole grain intake as part of a healthy diet to prevent type 2 diabetes [36]. Higher intake of vegetables was also a key component of this dietary pattern. Results from the Guangzhou Nutrition and Health Study (GNHS) showed that vegetables and gut microbiota diversity and composition were not associated with the risk of developing T2DM [37]. It may be related to the Chinese food culture, where most vegetables are cooked before consumption. One study showed that a higher intake of raw vegetables (rather than cooked vegetables) was positively associated with a lower risk of CVD [38]. Furthermore, the results of meta-analyses showed that a total vegetable intake of about 100 g per day did not affect the risk of developing T2DM. Increasing intake to 300 g per day could reduce the risk of developing T2DM by 9% with a non-linear dose-response association, whereas above this value, no significant benefit of increasing intake was observed [10,39]. Therefore, it is significant to investigate the potential different associations of raw vegetables and cooked vegetables and different intakes with T2DM in future work. Furthermore, high intakes of other meat products such as fish and poultry were also observed in this dietary pattern. However, a meta-analysis of prospective cohort studies found that these two food groups did not appear to be strongly associated with diabetes [10,40]. Finally, the intake of condiments such as salt was considered in the factor analysis and represented modern dietary patterns with a high factor loading. The proinflammatory response has an essential effect on the development of T2DM [41]. It has been shown that increased salt (sodium chloride) intake appeared to affect T2DM by enhancing TH17 cell activity through the p38/MAPK pathway and serum/glucocorticoid-regulated kinase 1 (SGK1) to increase proinflammatory cytokine levels [42]. In addition, a Japanese cohort study found that high HbA1C and dietary sodium intake had a synergistic effect, which increased the risk of CVD in patients with T2DM [43].

The fruit–milk dietary pattern, which is high in milk and its products, fruits, eggs, nuts and seeds and pastry snacks, but is low in vegetable oils, had no association with high blood glucose. This nonsignificant relationship might be the result of the interaction of certain healthy and unhealthy food groups. On the one hand, milk and its products, fruits and nuts and seeds as healthy foods may reduce the risk of developing DM. A prospective study from Singapore showed a significant 12% reduction in the risk of T2DM in daily milk drinkers compared to non-milk drinkers [44]. A meta-analysis including 14 cohort studies found a non-linear negative association of total dairy and low-fat dairy consumption with T2DM risk, with the inverse association appearing to be strongest at an intake of 200 g/day [39,45]. As the intake increased to 400–600 g/day, the risk was reduced by 6%. There was no significant benefit for increasing intake above this value [10]. In addition, total fruits, nuts and seeds were negatively associated with the risk of developing T2DM [10,37,46,47]. On the other hand, a higher intake of eggs and pastry snacks and a lower intake of vegetable oils were thought to be positively associated with T2DM [10,39]. Although this dietary pattern did not show a significant association with high blood glucose in this study, we need to recognise the drawbacks of this dietary pattern. For example, the huge market for snack foods or ultra-processed foods in China could cause a dramatic shock to the traditional Chinese diet.

To our knowledge, this is the first study to combine SEqM and multivariate logistic regression to examine the association between dietary patterns and high blood glucose in Jiangsu province, China. Moreover, our population was sampled according to strict criteria, with the results being representative. However, the present investigation has some shortcomings. First, the cross-sectional study design naturally hinders the inference of causality. Second, the data of the diet was chosen from a 3-day, 24-h dietary recall and weighing method, and seasonal factors may have led to biased food choices; third, other confounding factors, such as physical activity and sleep duration, were not considered in this study.

## 5. Conclusions

This study finally identified two dietary patterns through EFA and CFA: the modern dietary pattern and the fruit–milk dietary pattern. The modern dietary pattern characterised by a high intake of red meats was positively associated with high blood glucose among adults in Jiangsu province of China, while the fruit–milk dietary pattern was not significantly associated with high blood glucose.

## Figures and Tables

**Figure 1 nutrients-14-04111-f001:**
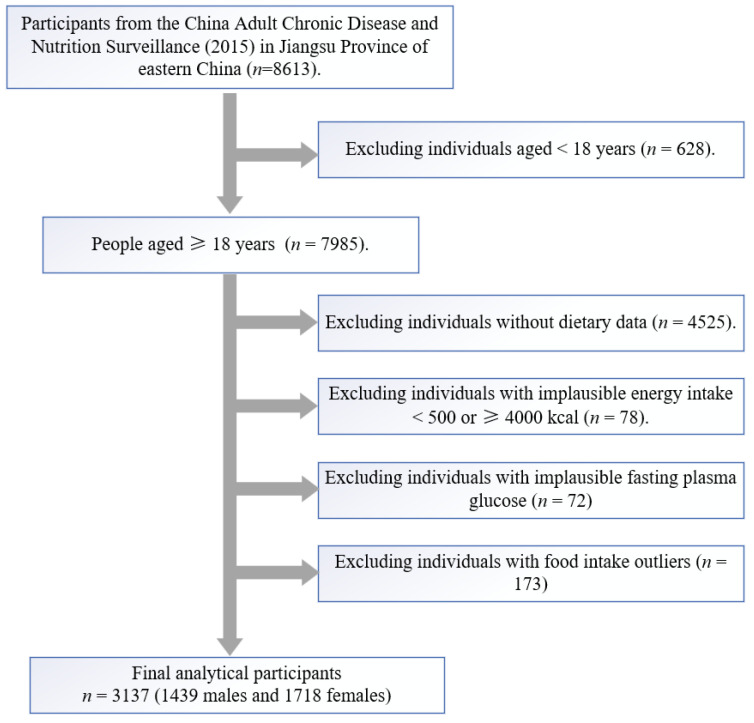
Flow chart of participants included in the study.

**Figure 2 nutrients-14-04111-f002:**
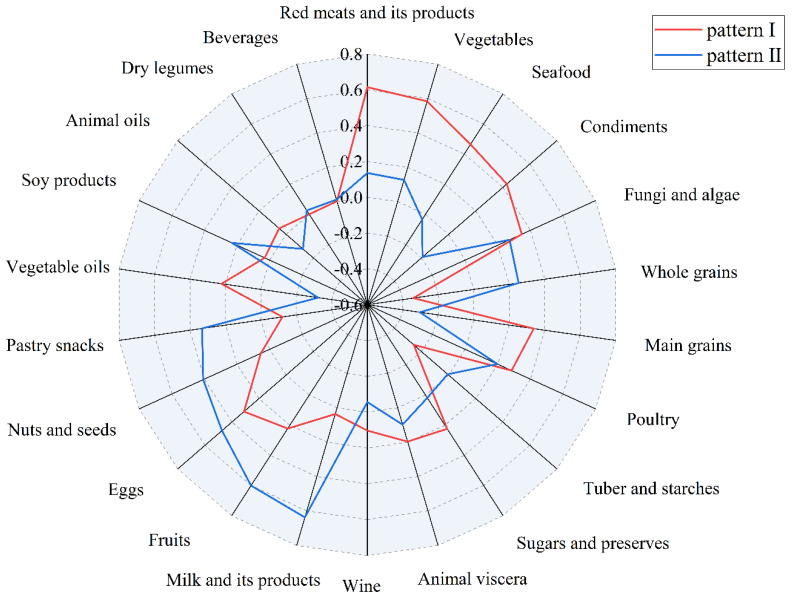
Radar plot of two dietary patterns by EFA.

**Figure 3 nutrients-14-04111-f003:**
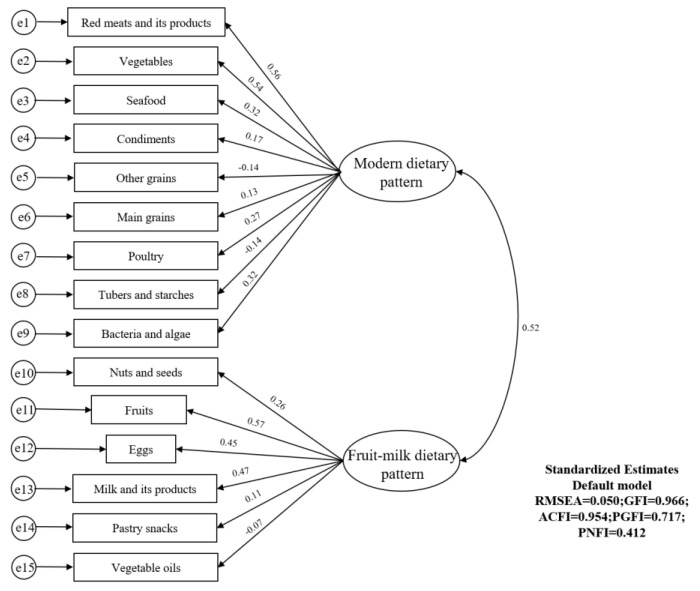
Measurement model for two dietary patterns by CFA. RMSEA = 0.050, GFI = 0.966, ACFI = 0.954, PGFI = 0.717 and PNFI = 0.412. e, error.

**Figure 4 nutrients-14-04111-f004:**
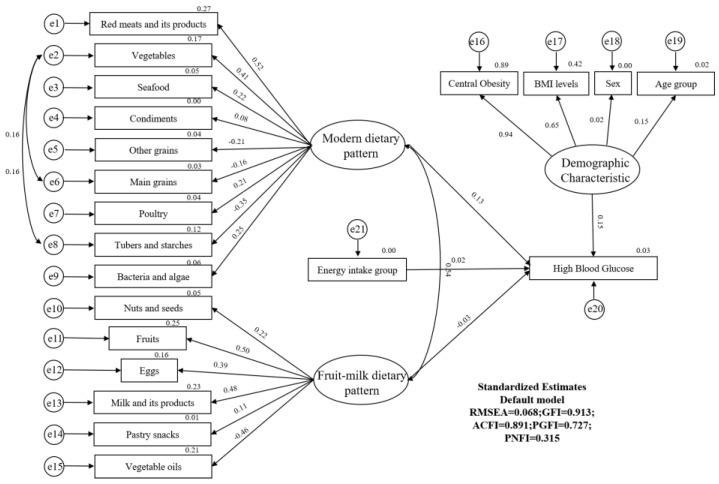
Final structural models. The path standardised coefficients of variables are presented on pathways. RMSEA = 0.068, GFI =0.913, ACFI = 0.891, PGFI= 0.727 and PNFI = 0.315. e, error.

**Table 1 nutrients-14-04111-t001:** Characteristics of the basic information distribution by gender.

Groups	Male		Female		*p*-Value
	Mean	SD	Mean	SD	
Age (years)	56.5	14.5	54.3	14.6	<0.001
Blood glucose (mmol/L)	5.5	1.4	5.4	1.3	0.031
BMI (kg/m^2^)	24.9	3.3	24.8	3.6	0.091
Energy intake (kcal/d)	1847.5	540.4	1540.8	440.9	<0.001
	*n*	%	*n*	%	*p*-value
Age group (years)					<0.001
18~34	143	10.0	214	12.5	
35~49	269	18.8	375	22.0	
50~64	542	37.9	671	39.3	
65~	475	33.2	448	26.2	
BMI level					<0.001
Thinness	22	1.5	38	2.2	
Normal	528	36.9	703	41.2	
Overweight	635	44.4	667	39.1	
Obesity	244	17.1	300	17.6	
Central obesity					0.574
No	865	60.5	1017	59.5	
Yes	564	39.5	691	40.5	
Smoking behaviour					<0.001
No	776	54.3	1684	98.6	
Yes	653	45.7	24	1.4	
Diabetes					0.223
No	1296	90.7	1570	91.9	
Yes	133	9.3	138	8.1	

**Table 2 nutrients-14-04111-t002:** Odds ratios (95% confidence intervals) for high blood glucose across quartiles of dietary patterns.

Groups	OR	95% CI	*p*-Value	*p* for Trend
Modern Dietary Pattern				
Q1	1.000			0.021
Q2	1.441	(0.992~2.094)	0.055	
Q3	1.566	(1.063~2.308)	0.023	
Q4	1.561	(1.025~2.379)	0.038	
**Fruit–milk dietary pattern**				
Q1	1.000			0.232
Q2	0.998	(0.687~1.451)	0.992	
Q3	1.060	(0.734~1.531)	0.755	
Q4	1.269	(0.887~1.814)	0.192	

**Table 3 nutrients-14-04111-t003:** Parameter Estimates from the SEqM of dietary patterns and high blood glucose among adults.

Path Analysis	Non-Standardised Coefficient	Standardised Coefficients	S.E.	C.R.	*p-*Value
Modern dietary pattern →diabetes	0.001	0.127	0.000	3.417	<0.001
Fruit-milk dietary pattern→diabetes	−0.003	−0.032	0.003	−0.903	0.366

## Data Availability

The datasets used and/or analyzed during the current study are available from the corresponding author on reasonable request.

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
