# Peer review of "The Relationship between Dietary Patterns and High Blood Glucose among Adults Based on Structural Equation Modelling"

_nutrients, 2022, doi:10.3390/nu14194111_

Round 1

Reviewer 1 Report

The topic of this paper is of considerable interest and the mathematical methods are of high value to identify how dietary patterns of Jiangsu Province may relate to risk for Diabetes.  There are, however, significant issues with the current version of the manuscript and will require re-analysis before it is acceptable for publication.

In the abstract and elsewhere using SEM as the abbreviation for the Structural Equation Modeling is confusing as SEM is commonly used for Standard Error of the Mean.  Suggest changing throughout to SEqM or other abbreviation that is different from SEM.

Lines 19-20.  English is not written by a native speaker:

Results: The 3137 participants with complete data were finally included in this study. The prevalence of diabetes was 9.3% and 8.1% in male and female, respectively, with no significant difference.  Should be “After exclusion 3137 participants with compete date were included in this study.  The prevalence of diabetes was 9.3% and 8.1% in males and females respectively.  There was no significant difference in diabetes prevalence between sexes.”

These types of errors (no agreement of singular and plural, incorrect phrasing etc.) are present throughout the manuscript and require significant editing.  Suggest use of one of the services that are recommended for these types of changes.

The introduction needs to better describe how the dietary patterns were chosen.

Line 11-114  States “Professionally trained enumerators record at the first household visit what foods the respondents have eaten in the past 24 hours and teach them how to record their intake”   Did the professional actually Visit the home of the participant?  How did they determine what foods were consumed?  Food frequency questionnaires?  Was food waste determined?  Did the Professional observe a meal?  This needs to be clarified and expanded.

Lines 123ff:  The categories used are where the major deficiency of this paper is revealed:

Wheat and rice can contain whole grains or for wheat be 100% whole grain this needs to be recategorized and reassessed

The category of “wine” lists beer and wine – this should be changed to “alcoholic beverages”  and any hard liquor should be included

100% juice should not be categorized with sugar sweetened beverages.  I do not know if artificially sweetened beverages are normally consumed by this population but if so they should not be categorized in the same manner as sugar sweetened beverages.

Flavored milk containing sugar should be analyzed separately than other milk products

What is the justification for categorizing animal viscera separately from other meat products many have essentially the same nutritional profiles.

Then entire table needs to be explained as to how these categories were defined (food Pagoda?  WHO definitions, other nutrient profiling system?

Cannot evaluate the results until the issues of how the food categories are determined is resolved

Author Response

The topic of this paper is of considerable interest and the mathematical methods are of high value to identify how dietary patterns of Jiangsu Province may relate to risk for Diabetes.  There are, however, significant issues with the current version of the manuscript and will require re-analysis before it is acceptable for publication.

  1. In the abstract and elsewhere using SEM as the abbreviation for the Structural Equation Modeling is confusing as SEM is commonly used for Standard Error of the Mean.  Suggest changing throughout to SEqM or other abbreviation that is different from SEM.

Answer: All abbreviation for structural equation models in the article have been changed to “SEqM”, which can be seen in red font.

  1. Lines 19-20.  English is not written by a native speaker:

Results: The 3137 participants with complete data were finally included in this study. The prevalence of diabetes was 9.3% and 8.1% in male and female, respectively, with no significant difference.  Should be “After exclusion 3137 participants with compete date were included in this study.  The prevalence of diabetes was 9.3% and 8.1% in males and females respectively.  There was no significant difference in diabetes prevalence between sexes.”

Answer: Thank you, professor. Lines 24-25 have been revised, which can be seen in red font.

  1. These types of errors (no agreement of singular and plural, incorrect phrasing etc.) are present throughout the manuscript and require significant editing.  Suggest use of one of the services that are recommended for these types of changes.

Answer: Thank you, professor. We have revised the whole draft and corrected the errors.

  1. The introduction needs to better describe how the dietary patterns were chosen.

Answer: We have described the selection of dietary patterns specifically in the methods section, which can be seen in lines 168-182.

  1. Line 11-114  States “Professionally trained enumerators record at the first household visit what foods the respondents have eaten in the past 24 hours and teach them how to record their intake”   Did the professional actually Visit the home of the participant?  How did they determine what foods were consumed?  Food frequency questionnaires?  Was food waste determined?  Did the Professional observe a meal?  This needs to be clarified and expanded.

Answer: Thank you, professor. The investigators not only recorded the 3-day 24-hour dietary intake questionnaire of the respondents but also weighed various condiments, so the investigators needed to visit the homes of the respondents for face-to-face interviews. The amount of food consumed was obtained by recall. The respondents were only required to recall what they had eaten, without considering food waste. As for the relationship with the food frequency questionnaire, this originally belongs to two different dietary survey methods.

  1. Lines 123ff:  The categories used are where the major deficiency of this paper is revealed:

Wheat and rice can contain whole grains or for wheat be 100% whole grain this needs to be recategorized and reassessed

Answer: Thank you, professor. According to your comments, we reconsidered the classification of food groups. We have changed the “whole grains” to “other grains”.

  1. The category of “wine” lists beer and wine – this should be changed to “alcoholic beverages” and any hard liquor should be included

Answer: Thank you, professor. We have changed the “wine” to “alcoholic beverages”.

  1. 100% juice should not be categorized with sugar sweetened beverages.  I do not know if artificially sweetened beverages are normally consumed by this population but if so they should not be categorized in the same manner as sugar sweetened beverages.

Answer: Thank you, professor. The grouping of “beverages” was determined according to the Chinese food composition table (2002), which can be seen in lines 163-164.

  1. Flavored milk containing sugar should be analyzed separately than other milk products

Answer: Thank you, professor. The grouping of “Milk and its products” was determined according to the Chinese food composition table (2002). I was also inspired by your idea about sugary flavored milk and other dairy products, sugary beverages and 100% juices, which can be used as a classification when we study dairy products or beverages separately at a later stage.

  1. What is the justification for categorizing animal viscera separately from other meat products many have essentially the same nutritional profiles.

Answer: When interviewing the survey respondents, we found that they generally consumed animal viscera and other meat products separately. The same grouping was carried out in other analyses on dietary patterns of Chinese populations, such as Jiguo Zhang, et.al(1), Jing Yan, et.al(2)

Reference:

(1). Zhang J, Wang H, Wang Y, Xue H, Wang Z, Du W, Su C, Zhang J, Jiang H, Zhai F, Zhang B. Dietary patterns and their associations with childhood obesity in China. Br J Nutr. 2015;113(12):1978-84.

(2) Yan J, Ren Q, Lin H, Liu Q, Fu J, Sun C, Li W, Ma F, Zhu Y, Li Z, Zhang G, Du Y, Liu H, Zhang X, Chen Y, Wang G, Huang G. Association between Dietary Patterns and the Risk of Depressive Symptoms in the Older Adults in Rural China. Nutrients. 2022;14(17):3538.

  1. Then entire table needs to be explained as to how these categories were defined (food Pagoda?  WHO definitions, other nutrient profiling system?

Answer: These food groupings were defined according to Chinese food composition table (2002), which can be seen in lines 163-164.

Reviewer 2 Report

The paper by Yuanyuan Wang describes the modern dietary patterns in Jiangsu Province, China increased a risk of diabetes using the structural equation model. Especially high intake of red meats in the modern dietary patterns was positively associated with diabetes in the adult population. The authors have some novel data but some of the methods need to be clarified

1.     What is Q1-Q4 in table 3

2.     Where is Supplementary Table 1 (3.2 section)?

3.     Please add space in wassignificantly (3.1. section last sentence).

4.     Please define several abbreviations (KMO and CVD) in text.

5.    Please correct reference number. Each reference has double number.

Author Response

The paper by Yuanyuan Wang describes the modern dietary patterns in Jiangsu Province, China increased a risk of diabetes using the structural equation model. Especially high intake of red meats in the modern dietary patterns was positively associated with diabetes in the adult population. The authors have some novel data but some of the methods need to be clarified

  1. What is Q1-Q4 in table 3

Answer: Q1-Q4 were the factor scores that we divided into quartiles for logistic regression analysis, which can be seen in lines 212-214.

  1. Where is Supplementary Table 1 (3.2 section)?

Answer: Supplementary Table 2 mainly showed the factor loadings (|>0.25|) for each food group. We have re-uploaded Supplementary Table 2.

  1. Please add space in wassignificantly (3.1. section last sentence).

Answer: Thank you, professor. We have added space in wassignificantly.

  1. Please define several abbreviations (KMO and CVD) in text.

Answer: We have defined KMO and CVD in lines 172 and 78, which can be seen in red font.

  1. Please correct reference number. Each reference has double number.

Answer: Thank you, professor. We have revised the reference number.

Reviewer 3 Report

Wang et al. performed an analysis to 1) derive the diet patterns among 3137 participants from the 2015 Chinese Adult Chronic Disease and Nutrition Surveillance Program using factor analysis; and 2) investigate the derived diet patterns with type 2 diabetes.

1.      Page 2, second paragraph, second sentence doesn’t make much sense. Factor analysis is used for grouping purposes, while SEM model seeks to solve the structural relationship, or more commonly, used for causality. Does the author aim to explore the causality? Or just association?

2.      With the blood samples collected, it would be helpful to use HbA1c to further determine diabetes diagnose.

3.      The authors may consider focusing on type 2 diabetes (T2D) only or conducting a sensitivity analysis among T2D, because T2D is more closely related to lifestyle factors.

4.      Table 3, is there any trend across the Q1-Q4? Does the relationship between diet patterns depend on quantiles?

5.      The language needs to be refined. Some words are weird, e.g., in the abstract, bacteria and algae…what does it mean by bacteria in diet pattern?

Author Response

Wang et al. performed an analysis to 1) derive the diet patterns among 3137 participants from the 2015 Chinese Adult Chronic Disease and Nutrition Surveillance Program using factor analysis; and 2) investigate the derived diet patterns with type 2 diabetes.

  1. Page 2, second paragraph, second sentence doesn’t make much sense. Factor analysis is used for grouping purposes, while SEM model seeks to solve the structural relationship, or more commonly, used for causality. Does the author aim to explore the causality? Or just association?

Answer: In this manuscript, our aim was to examine the association between dietary patterns and diabetes through the structural equation modeling study. The second sentence was revised and can be seen in lines 83-85.

  1. With the blood samples collected, it would be helpful to use HbA1c to further determine diabetes diagnose.

Answer: Indeed, in our survey, only fasting blood glucose values were tested, which can be used alone for diabetes epidemiological surveys. We changed diabetes to high blood glucose.

  1. The authors may consider focusing on type 2 diabetes (T2D) only or conducting a sensitivity analysis among T2D, because T2D is more closely related to lifestyle factors.

Answer: Considering the point you raised, we define fasting blood glucose values above 7.0 as high blood glucose, rather than as diabetes.

  1. Table 3, is there any trend across the Q1-Q4? Does the relationship between diet patterns depend on quantiles?

Answer: Yes. The results of multifactorial logistic regression showed that modern dietary patterns were associated with an increased risk of diabetes and showed a trend toward elevation with increasing intake. We revised the results and table of this section, which can be seen in lines 272-277.

  1. The language needs to be refined. Some words are weird, e.g., in the abstract, bacteria and algae…what does it mean by bacteria in diet pattern?

Answer: Thank you, professor. We have refined the language of the manuscript.

Round 2

Reviewer 1 Report

The manuscript is much clearer and has been greatly improved.  There are still some English style issues but this can be handled by the editorial office.

The methods are now much more clear as well and the basis for the categorizations are explained.

Reviewer 2 Report

The authors have made substantial revisions in response to the original critique.

Reviewer 3 Report

I have no further comments.